# Research on Spectrum Prediction Technology Based on B-LTF

Xue Wang , Qian Chen and Xiaoyang Yu *

School of Measurement and Communication Engineering, Harbin University of Science and Technology, Harbin 150080, China
* Correspondence: 2220600068@stu.hrbust.edu.cn

**Abstract:** With the rapid development of global communication technology, the problem of scarce spectrum resources has become increasingly prominent. In order to alleviate the problem of frequency use, rationally use limited spectrum resources and improve frequency utilization, spectrum prediction technology has emerged. Through the effective prediction of spectrum usage, the number of subsequent spectrum sensing processes can be slowed down, and the accuracy of spectrum decisions can be increased to improve the response speed of the whole cognitive radio technology. The rise of deep learning has brought changes to traditional spectrum predicting algorithms. This paper proposes a spectrum predicting method called Back Propagation-Long short-term memory Time Forecasting (B-LTF) by using Back Propagation-Long Short-term Memory (BP-LSTM) network model. According to the historical spectrum data, the future spectrum trend and the channel state of the future time node are predicted. The purpose of our research is to achieve dynamic spectrum access by improving the accuracy of spectrum prediction and better assisting cognitive radio technology. By comparing with BP, LSTM and Gate Recurrent Unit (GRU) network models, we clarify that the improved model of recurrent time network can deal with time series more effectively. The simulation results show that the proposed model has better prediction performance, and the change in time series length has a significant impact on the prediction accuracy of the deep learning model.

**Keywords:** cognitive radio (CR); spectrum prediction; deep learning; long short-term memory network (LSTM); long sequence time-series forecasting (LSTF)



## 1. Introduction

Information has changed human life, from sharing bicycles to sharing everything, from the commercialization of 5G to the research and development of 6G. In today's era, communication technology is vigorously developed, and the low-Earth orbit satellite Internet of things is used as the architecture to create a blue ocean of satellite Internet of things applications to realize the Internet of everything. With a large number of devices being connected to the wireless network, the scarcity of wireless spectrum resources is becoming increasingly serious, and the whole wireless communication environment has become extremely severe. The current spectrum resource allocation strategy is mainly a static spectrum allocation strategy, which licenses the spectrum to a limited number of users. The decision stipulates that even if the licensed band is idle, other users are still not able to access and use it, which results in a shortage of spectrum resources, but it is a lot of waste.

According to the research report of the Federal Communications Commission (FCC) of the United States, the spectrum occupancy of different licensed frequency bands varies due to the different number of services and requirements, ranging from 15% to 85% [1]. Additionally, according to the survey of global spectrum utilization in recent years, the spectrum utilization rate is very low, and there is a large amount of spectrum waste worldwide. The American University of Sharjah measured the local UHF band (300 MHz to 3000 MHz) in 2017, and the study showed that the average occupancy rate was between 10% and 35% [2]. Yu et al. used R&S EM100 digital compact receiver to measure the

spectrum occupancy of China's radio broadcast allocated frequency band from 87.5 MHz to 108 MHz, and the results showed that there were a large number of spectrum holes [3]. In 2018, spectrum measurements of 6 GHz band models at three different locations in San Luis Potosi, Mexico, showed that the spectrum utilization rate was only 4.73% [4]. In 2019, using RF Explorer 6G Combo to detect the spectrum occupancy of the GSM900 band in the Samsun area, and the calculation results showed that the occupancy rate was about 8.5% and 82% at −40 dBm and −75 dBm, respectively [5]. The 24 h detection was carried out on the frequency band from 30 MHz to 1030 MHz in Pakistan, and the results showed that the average spectrum utilization rate was only 35.41%, among which the minimum occupancy rate of the 850 MHz to 1030 MHz band was only 25% [6]. As can be seen in the above data, spectrums in large Numbers in the world cannot be effectively used, according to research, at any time and any place, the average spectrum utilization rate is not more than 5% [7].

As early as 1999, in order to solve the problem of spectrum resources, Dr. Joseph Mitola proposed the concept of Cognitive Radio (CR) [8]. After several years of radio technology development, the current CR technology mainly includes spectrum sensing, spectrum decision, spectrum shifting and spectrum sharing [9–11]. Through the above four modules, CR technology has largely improved the spectrum resources problems. According to the data of "Cognitive Radio Market Report", CR technology can effectively improve spectrum utilization, reaching 3–10% [12]. Spectrum sensing technology is an indispensable part of the four modules, which realizes the dynamic access of the spectrum by sensing the holes in the spectrum environment. However, in the process of spectrum sensing, in order to obtain more spectrum resources, the range of spectrum sensing is often increased, but the corresponding perception time will be extended, and a lot of calculations and more and more complex algorithm processes will be generated. In order to solve such problems, Haykin first proposed spectrum prediction technology based on spectrum sensing technology in 2005, and then around 2012, researchers strengthened the concept of spectrum prediction technology. First of all, spectrum prediction technology can find the relationship between the spectrum through the study of the historical spectrum segment, predict the future spectrum state, find a range of possible free frequency bands, and then carry out spectrum perception of this range, which can directly find the free spectrum resources in a short time, so as to solve the problem of high spectrum perception consumption. Secondly, when the received signal is incomplete, we can also predict the amplitude and trend of the complete signal according to the spectrum prediction technology and carry out information demodulation. Finally, it allows users to access the underutilized spectrum opportunistically in the time domain or spectrum domain without significantly affecting other users, which provides the possibility for subsequent spectrum decision making, spectrum migration and spectrum sharing, thus assisting cognitive radio technology.

In the current research, the prediction results of spectrum prediction technology are relatively single, which can only complete the prediction in the ordinary time dimension but cannot reflect the characteristics of the frequency band at that time. The existing spectrum prediction processes can be roughly divided into two categories [13], namely, by predicting the quality of the channel and predicting from the perspective of channel occupancy. The current common spectrum prediction technology methods are the regression analysis Model and Hidden Markov Model (HMM). The model based on HMM is simpler and widely used in CR. This kind of method has better accuracy in predicting spectrum occupancy even in complex industrial environments [14]. However, the traditional HMM spectrum prediction algorithm has the problems of time extension, and the prediction accuracy is easily affected by the uncertainty of the matrix to be tested. For this reason, [15] proposes a spectrum occupancy prediction method based on a segmented Markov model, which uses a density clustering density algorithm to predict channel states in clusters. This algorithm not only improves the prediction accuracy but also reduces the prediction uncertainty. The piecewise prediction method of the periodic channel can better describe the change characteristics of historical data in the periodic channel and improve the final prediction accuracy without

changing the algorithm complexity [16]. The statistical learning method is used to learn the historical data of spectrum perception. Under the condition of ensuring the prediction rate, based on HMM, the optimal frequency of shortwave cognition is found to accurately predict the channel state of shortwave users [17]. In order to solve the shortcomings of the existing channel access technology in the shortwave ALE system, the three-state HMM is proposed to be used in spectrum prediction technology in [18], which greatly improves the prediction accuracy and utilization rate in the shortwave ALE system. Due to the nature of these two algorithms in mathematical calculation, although it will be convenient to understand the prediction process, in general, in the spectrum occupancy prediction method based on HMM, the system consumption is relatively large. Furthermore, the regression method is highly complex, and is not suitable for continuous prediction. In addition, the idea of cooperative prediction can also be applied to prediction technology. For the energy-constrained cognitive radio network, the combination of the cooperative prediction model and the spectrum sensing framework can overcome the problems of the local prediction model, so as to improve the spectrum efficiency in the case of low energy consumption [19].

At present, these prediction techniques solve the spectrum problem by developing complex algorithms, which often consume a lot of time and energy in the prediction, which hinders the success of dynamic spectrum access. Therefore, we urgently need flexible and intelligent spectrum prediction technology. The emergence of machine learning brings new hope to many fields, among which deep learning has shown good ability in the classification and prediction of some complex systems, giving researchers the idea of applying deep learning to spectrum prediction technology.

In [20], a spectrum prediction technology based on Multilayer Perceptron (MLP) is proposed. The structural parameters of the artificial neural network are configured, and all parameters are randomly initialized and then iteratively trained. The gradient is constantly calculated, and the parameter values are updated until the iteration conditions are met. However, due to the structure of the MLP network itself, the learning speed is slow, and it is easy to fall into the situation of local extremum. The complex nonlinear processing ability of feedforward neural networks can greatly improve the learning speed and prediction accuracy in prediction, among which the BP network [21] and its variants are common. Later, researchers proposed Recurrent Neural Networks (RNN) on the basis of BP, which were constructed by sliding windows to analyze spectrum data in a specified time. However, RNN is prone to the problem of gradient disappearance in spectrum prediction, so a spectrum prediction method based on the Deep Recurrent Neural Network (DRRN) [22] and a series of improved network algorithms are proposed. In [23], spectrum prediction techniques based on the Long Short-term Memory network (LSTM) and BP network were discussed. In [24], a new spectrum prediction framework was developed for LSTM, which improved its processing of time series. In [25], prediction errors of LSTM were compared with those of integrated moving average autoregression and delay neural networks, which proved that LSTM has certain advantages and good prediction performance in time series problems. In order to solve the long-term predictions of spectrum data, ref. [26] proposes a convolutional LSTM model for spectrum prediction—ConvLSTM. In the 450–520 MHz frequency band, the long-term spatial-spectral-time joint prediction of the spectrum signal is carried out. The results show that the model exhibits a stable value of Root Mean Squared Error (RMSE) for different channels, but it rises with the increase in the time step. In [27], STS-PredNet was proposed by combining ConvLSTM and the predictive recurrent neural network (PredRNN). This model shows stable RMSE performance in multi-time step spatial-temporal spectrum prediction, and effectively improves the prediction performance in different prediction ranges. For the ISM band with high burst characteristics, ref. [28] proposed an algorithm named Classified-based Deep Reinforcement Learning (C-DRL) to quickly and efficiently complete the matching between the state and the prediction model, achieving the effect of fast convergence, simple operation and high prediction rate. In order to reduce the time cost of convolutional network prediction in multi-channel, Gao et al. use

LSTM, Seq-to-Seq modeling, exploit the intrinsic correlation and time correlation between channels, and propose a multi-channel multi-step joint spectrum prediction algorithm, which can effectively improve the prediction accuracy of multi-channel [29].

In addition, for different industrial frequency bands and different communication systems, researchers have also proposed other prediction algorithms based on LSTM networks, which contribute greatly to the research of spectrum prediction technology [30–34]. Despite the high performance of LSTM, it may bring challenges in terms of computational burden and handling missing data, and since spectral data is equivalent to long time series and LSTM has reduced prediction accuracy for long sequences, we propose the B-LTF algorithm to improve the problem of long sequence prediction. It slows down the impact of sequence length change on the prediction performance of the network so that the overall spectrum prediction rate is improved. Secondly, in this paper, we not only predict the channel state but also predict the spectrum signal. The prediction of the channel state is first processed by the gate threshold and then predicted, so as to know the channel state of the node in the future time, and the signal spectrum prediction goes deeper into the deeper level, that is, the trend and amplitude change of the future signal can be known through the sparse spectrum signal data, which is of great help to the subsequent cognitive radio technology.

Since deep learning algorithms are typically exploratory, they do not have any preconditions or assumptions about the data. Therefore, in many cases, they provide higher accuracy than traditional probabilistic and statistical algorithms. Although deep learning has been very popular in recent years, it is still in the initial stage in the field of communication, especially in CR technology. Therefore, how to use deep learning algorithms to make spectrum prediction more efficient and effective is an emerging research direction. Our main contributions in this paper are summarized as follows:

- We study the problem of spectrum availability prediction and discuss the time spectrum occupancy characteristics together.
- We proposed the B-LTF algorithm, combined the BP network with LSTM, built a new network structure and realized the spectrum prediction from the neural network. We studied the influence of the sequence length of spectrum data and the model prediction rate in detail and effectively improved the accuracy of spectrum prediction.
- We show that the analysis of simulation prediction values obtained by simulating the current channel state shows that a long short-term memory network and its improved model can effectively process the time series, the GRU model has a simpler structure and the training time of the GRU model will be significantly reduced compared with the LSTM network, and the improved B-LTF algorithm compared with the LSTM, BP and GRU has better predictive performance. In addition, when the sequence length of spectrum data increases, the model prediction rate tends to be saturated or reduced.

The rest of this paper is organized as follows. We first introduce the spectrum prediction models, deep learning models and related knowledge in Section 2. Section 3 briefly describes the designed B-LTF model for spectrum prediction, which is followed by simulation results and discussions in Section 4. Finally, we conclude the paper in Section 5.

## 2. System Model

### 2.1. Spectrum Prediction Model

In CRN, spectrum sensing techniques work by sensing the frequency band of interest in time slot units, where the time length of each time slot is $T_s$. The sensed signal strength $\mu(t)$ of each time slot is compared with a gate threshold $\lambda$, and whether this frequency band is occupied is determined by the following decision rule.

$$\gamma(t) = \begin{cases} 0 & \mu(t) < \lambda \\ 1 & \mu(t) \geq \lambda \end{cases} \tag{1}$$



The number "1" indicates that the frequency band is occupied by the PU, and "0" indicates that the frequency band is not occupied. As shown in Figure 1, red indicates that the frequency band is occupied, and white indicates that the frequency band is idle:

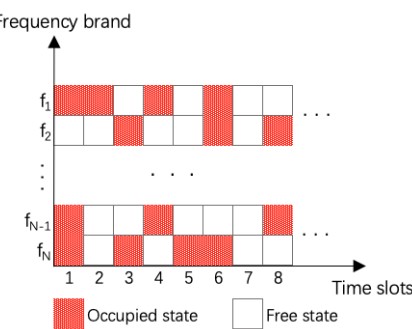

**Figure 1.** Analog frequency band usage.

After sensing the frequency band, the spectrum prediction technology uses the time-dependent spectrum point data for prediction and predicts the spectrum state of the next time slot by mining the internal relationship between the spectrum states of the historical time slot. In other words, assuming that there are T historical time slots of spectral data, through $X_{t-T+1}$, $X_{t-T+2}$, $\cdots\cdots$, $X_t$ performs network training to find internal connections so as to obtain the predicted value of $X_{t+1}$ for time slot $t + 1$. The detailed prediction model is shown in Figure 2.

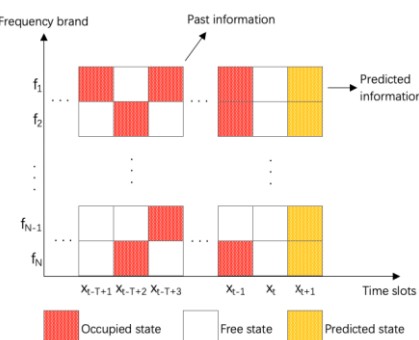

**Figure 2.** An example of a spectrum prediction model.

### 2.2. Deep Learning Model

In the current research of radio technology, the neural network is regarded as one of the effective methods to solve spectrum problems. It usually consists of an input layer, one or more hidden layers, and an output layer. By changing the number of hidden layers and the number of neurons in each layer, different complex nonlinear functions can be constructed. The current network types are mainly divided into feedforward neural networks and feedback neural network. The following mainly introduces the improved models of BP network and improvement of LSTM.

#### 2.2.1. Conventional BP Network

In 1986, the research team led by Rumelhart and MeCelland first proposed the BP neural network, which trains the feedforward network according to the error backpropagation algorithm and is one of the common models in the current neural network model. In terms of network structure, the model structure of the BP neural network is relatively simple and is mainly composed of three parts: input layer, hidden layer and output layer. Figure 3 shows the structure of a single-layer BP neural network.

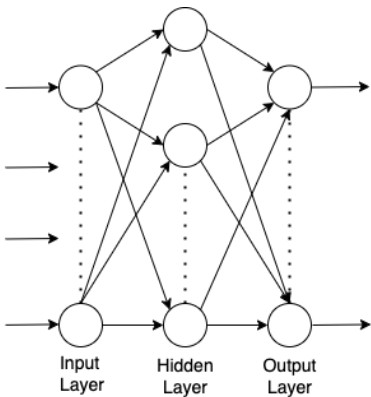

**Figure 3.** Single-layer BP neural network architecture.

The BP-neural network algorithm is as follows (Algorithm 1):

---

**Algorithm 1.** BP-neural network algorithm

---

1. Get the training data set XTrain and test data set XTest.
2. Set the structure parameters of BP neural network model.
3. Input XTrain, by passing forward: from input layer to hidden layer to output layer, get X_Train.
4. compare the XTrain with the YTrain to obtain the prediction error.
5. When the prediction error e > eth is satisfied and the number of iterations Nit < n, the error backward propagation process is performed to update the weights and then go back to 1, otherwise to 6.
6. When e < eth or Nit = n, the training is finished and obtain the trained network model.
7. Put the XTest into the trained BP neural network to obtain YTest.

---

where YTrain is the expected output, eth is the specified error size, and n is the set number of training sessions.

For the prediction model design of the BP network, we set three network layers: an input layer, an output layer and a hidden layer. Since a single hidden layer is enough to realize the nonlinear mapping relationship of any dimension, there is no need to increase the number of hidden layers but increase the training time. In addition, the number of nodes in the hidden layer is set to 288, and the number of neurons in the input layer and output layer is set to 1. It is specified that the maximum number of trainings is 250, the learning rate is 0.005, and the target error accuracy of training is 0.00001. The detailed BP network structure diagram is shown in Figure 4.

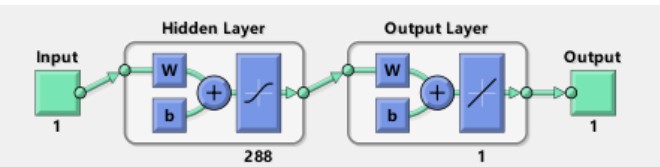

**Figure 4.** Designed BP network model.

2.2.2. Emerging LSTM Network

In order to solve the problems such as gradient disappearance in RNN, an LSTM network is designed by introducing a threshold mechanism, namely the memory unit [35]. Figure 5 shows the structure of the memory cell. It is mainly composed of memory cells and three gates: the input gate, forget gate and output gate.

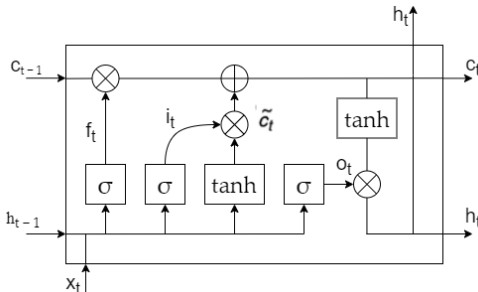

**Figure 5.** The memory cell structure of an LSTM network.

The LSTM network model designed in this paper will consist of five layers, first an input layer, then an LSTM layer, followed by a forgetting layer, immediately followed by a fully connected layer, and finally an output layer. The number of nodes in the LSTM layer is set to 1024, the activation function of the hidden layer uses tanh activation function, the initialization uses orthogonal initialization, the activation function of the full connection layer is set to the '*softmax*' activation function, and the forgetting rate is set to 0.2 in the forgetting layer. The optimizer of the whole LSTM network uses Adaptive momentum (Adams) algorithm optimizer, and the number of training sessions is set to 200–300. The detailed network structure diagram is shown in Figure 6. At present, the LSTM layer is set to one layer, but the number of LSTM layers may be increased in the subsequent simulation in order to explore the influencing factors of the prediction performance.

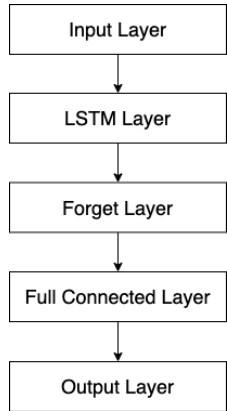

**Figure 6.** The designed LSTM network model.

### 2.2.3. GRU Network Model

GRU neural network is a popular neural network in recent years, which is a variant of the LSTM network. Compared with the three gate structures of the LSTM network, the network structure of GRU is simpler, with only two gate structures: the update gate and reset gate and the specific network structure is shown in Figure 7.

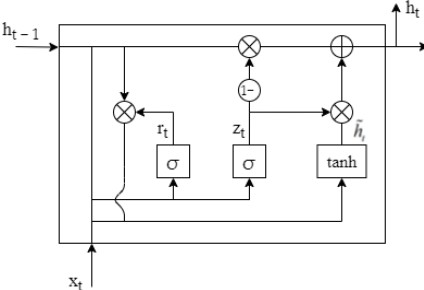

**Figure 7.** GRU network unit model.

The reset gate and update gate in the GRU network act as the function of updating and filtering the input signal [36]. The reset gate "$r$" controls the amount of the previous state signal flowing into the current state, and the update gate "$z$" controls the degree of the current state when the previous state signal flows into the current state. The specific mathematical formula is shown below.

$$r_t = \sigma(W_r \cdot [h_{t-1}, x_t]) \tag{2}$$

$$z_t = \sigma(W_z \cdot [h_{t-1}, x_t]) \tag{3}$$

$$\widetilde{h}_t = \tanh\big(W_{\widetilde{h}} \cdot [r_t \otimes h_{t-1}, x_t]\big) \tag{4}$$

$$h_t = (1 - z_t) \otimes h_{t-1} + z_t \otimes \widetilde{h}_t \tag{5}$$

where $\sigma$ is the *sigmoid* function, tanh is the hyperbolic tangent activation function, and $W$ is the weight matrix of the corresponding module. Equation (2) shows that when time step is $t$, the input vector is the result of linear transformation obtained by multiplying $x_t$ and the state $h_{t-1}$ saved at the previous time step $t-1$ with the weight matrix, respectively. The reset gate uses an activation function to convert the sum of two results into [0,1]. Equation (3) is the relevant formula of the update gate, which is similar to the expression of the reset gate. Equation (4) is expressed as the memory content in the current cell, and the current information and historical information are added and transformed by the tanh function. Equation (5) represents the final output of the memory unit.

From the structural model and calculation formula of the GRU network, it can be seen that the GRU network, such as the LSTM network, memorizes the information of historical time steps and will not be eliminated with the increase in time steps, but combines the historical state information with the current state information, and then passes it to the subsequent cells. However, compared with the LSTM network, when solving consistent types of problems, because the structure of GRU is simpler, the time required for the whole training will be greatly reduced, and the final results are similar to the results of the LSTM network.

The GRU network is also similar to the LSTM network model in model design. Firstly, a GRU layer is set, the number of nodes in the network layer is set to 1024, the tanh function is also used as the activation function, the optimizer chooses Adams, the number of training sessions is set to 200–300, and a fully connected layer follows the GRU layer. There is also an input layer and an output layer, and the complete GRU network model is shown in Figure 8.

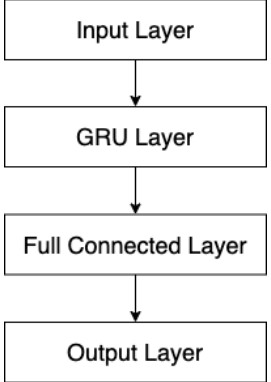

**Figure 8.** The designed GRU network model.

## 3. B-LTF Model

### 3.1. Related Theories and Formulas

Compared with the traditional HMM model, BP neural network can store a large number of specific connections between input and output, obtain the estimation error

of all levels through reverse transmission, and then use the steepest descent method to continuously update the weights and thresholds of the whole network model to achieve the purpose of minimizing the error. The process is mainly divided into two stages. The first stage is the forward propagation of the signal from the input layer through the hidden layer and finally to the output layer. The second stage is the back propagation of the error, from the output layer to the hidden layer, and finally to the input layer, adjusting the weights and biases from the hidden layer to the output layer in turn, and the weights and biases from the input layer to the hidden layer. The detailed mathematical formula is as follows:

Hidden layer output:

$$H_j = g\left(\sum_{i=1}^{n} \omega_{ij} x_i + a_j\right) \tag{6}$$

Output from the output layer:

$$O_k = \sum_{j=1}^{l} H_j \omega_{jk} + b_k \tag{7}$$

Error formula:

$$E = \frac{1}{2}\sum_{k=1}^{m}(Y_k - O_k)^2 \tag{8}$$

Weight update formula:

$$\begin{cases} \omega_{ij} = \omega_{ij} + \eta H_j(1 - H_j)x_i\sum_{k=1}^{m}\omega_{jk}e_k \\ \omega_{jk} = \omega_{jk} + \eta H_j e_k \end{cases} \tag{9}$$

Bias update formula:

$$\begin{cases} a_j = a_j + \eta H_j(1 - H_j)\sum_{k=1}^{m}\omega_{jk}e_k \\ b_k = b_k + \eta e_k \end{cases} \tag{10}$$

The number of nodes in the input layer is $n$, the number of nodes in the hidden layer is $l$, and the number of nodes in the output layer is m. The weights $\omega_{ij}$ from the input layer to the hidden layer, the weights $\omega_{jk}$ from the hidden layer to the output layer, the bias $a_j$ from the input layer to the hidden layer, and the bias $b_k$ from the hidden layer to the output layer. The learning rate is $\eta$ and the activation function is $g(x)$. The activation function is taken as a *Sigmoid* function.

The LSTM network can perform selective memory of historical information due to the existence of three gates and can better deal with time series problems. The input gate $i$ controls how much information goes into the memory cell, the forget gate $f$ determines how much state information of previous cells is remembered, the output gate $o$ controls the amount of information output by calculating the storage size of the memory cell, and the most critical memory cell gives the LSTM the ability to freely choose the content of memory in each time step. The mathematical model of the network is as follows:

$$i_t = \sigma\left(W^i x_t + U^i h_{t-1} + b_i\right) \tag{11}$$

$$f_t = \sigma\left(W^f x_t + U^f h_{t-1} + b_f\right) \tag{12}$$

$$o_t = \sigma(W^o x_t + U^o h_{t-1} + b_o) \tag{13}$$

$$\widetilde{c}_t = \tanh(W^c x_t + U^c h_{t-1} + b_c) \tag{14}$$

$$c_t = f_t \otimes c_{t-1} + i_t \otimes \widetilde{c}_t \tag{15}$$

$$h_t = o_t \otimes \tanh(c_t) \tag{16}$$

where $x_t$ is the input vector at time step t, the hidden layer state vector is $h_t$, $c_t$ is the memory cell, $\sigma$ is the sigmoid function, tanh is the hyperbolic tangent activation function, $W$ and $U$ are the weight matrix of the corresponding module, and $b$ is the bias vector of the corresponding template. Equations (11) and (14) are related calculations for input gates, Equation (15) is the cell state equation at time '$t$', Equation (12) is related calculations for the forget gate, Equations (13) and (16) are related calculations for the output gate.

### 3.2. Model Introduction

This paper mainly focuses on the prediction and research of spectrum signals. Since spectrum signals are equivalent to a long time series, their prediction is a series-to-sequence prediction. Therefore, according to this situation, we proposed the B-LTF algorithm, which is mainly a prediction model composed of the BP neural network and LSTM network. Among them, the BP neural network is not only simple in structure but also has strong nonlinear mapping ability. However, it does not consider the time correlation between the data, which often has a big problem for the prediction of time series. When facing the same data set, it will still wait for the same result after transformation. In addition, some traditional neural networks are prone to problems such as the disappearance of the gradient in the prediction, but the LSTM network changes this situation. It can solve the gradient problem in the prediction, and because of the existence of gates, the LSTM neural network has better performance in dealing with time series problems. Therefore, combining the advantages of the BP neural network and the LSTM neural network, the construction of the BP-LSTM combined network prediction model can not only solve the deficiencies of the BP model but also improve the convergence speed of the overall model, so as to improve the prediction rate of the model.

Before training, the data set was classified into the training set and prediction set. In the training, the training data is first preprocessed and then input into the BP neural network model and LSTM neural network model, respectively. After the BP neural network and LSTM neural network training, two trained network models are obtained, as shown in Figure 9 In the process of prediction, the data is first input into the BP network model to obtain the output result $y_1$, and then $y_1$ is input into the LSTM network model to obtain the output data $y_2$, and finally, $y_1$ and $y_2$ are weighted by the weight ratio to obtain the output data $y_3$, that is, the final prediction result. The complete process of prediction algorithm (Algorithm 2) and prediction model is as follows:

---

**Algorithm 2.** B-LTF algorithm

---

**Input:** Historical spectrum data {$X_1$, $X_2$, . . . , $X_t$}, Time dimension T;
**Output:** The spectrum prediction results;
//Construct the dataset;
1. The spectrum data is preprocessed according to the threshold value, so as to turn the data into a sequence composed of [0,1] "$S_p$" represents the frequency band state
2. $\mathcal{D} \leftarrow \varnothing$
3. Put the sample $S_p$ into $\mathcal{D}$
4. A segment is randomly selected in $\mathcal{D}$ and divided into $\mathcal{D}_{Train}$ and $\mathcal{D}_{Test}$ according to the ratio of training set to test set
//Train the model;
5. Initialize all learnable parameters W, b in BP-LSTM
6. **Repeat**
7.     Randomly select a batch of instances from $\mathcal{D}_{Train}$
8.     Update W, b in the network
9. **Until** the training epochs are met
//Test the model;
10. Put $\mathcal{D}_{Test}$ into the trained model
11. Output the prediction results

---

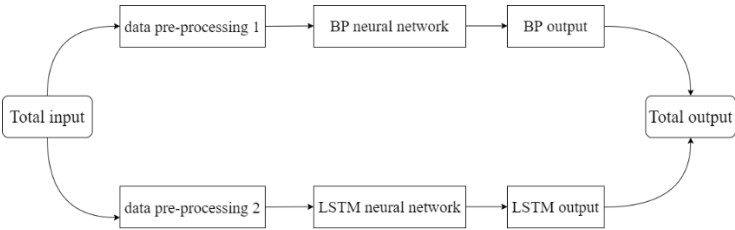

**Figure 9.** B-LTF training flow.

For the B-LTF network model, we set the training period epochs = 250, the optimizer adopts Adam. For B-LTF network parameters, it is divided into the BP layer and LSTM layer, the number of neurons is 288 and 1024, and the neuron loss rate is 0.2. In addition, different from the prediction using BP neural network alone, the output dimension of the BP layer needs to be consistent with the input dimension of the LSTM layer. The detailed model framework is shown in Figure 10.

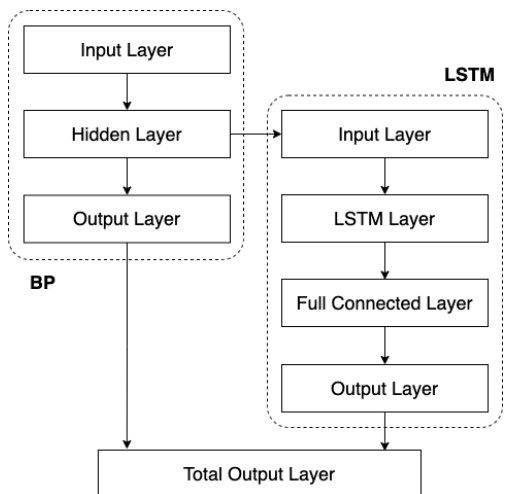

**Figure 10.** The designed B-LTF network model.

## 4. Experiment Evaluation

In this paper, we use the B-LTF network model and three common neural networks to make two kinds of simulation prediction. The first one is to compare the accuracy of each neural network for the future channel state prediction, change the respective network parameters, and explore the factors affecting the prediction degree. The second is sequence-to-sequence prediction, which predicts the trend and amplitude of the spectrum signal in the future period through the known spectrum signal.

### 4.1. Channel State Prediction

In this paper, the spectrum data are from the use of the high-performance spectrum analyzer AgilentE4440A by RWTH Aachen University in Germany on the wireless spectrum of a school in Maastricht, Netherlands. The measurement band range is mainly 20 MHz to 1.52 GHz and 1.5 GHz to 3 GHz. The whole measurement band is scanned each time, and the scanning time is 1.8 s on average. The measurement resolution bandwidth is 200 KHz. Because of the huge number of spectrum data measured, it is not conducive to the simulation of all the spectrum data, so we selected the frequency band data from 890 MHz to 915 MHz (GSM900 uplink band) and 935 MHz to 960 MHz (GSM900 downlink band) from the collected spectrum data for processing and data analysis. Then, nine channels from the downlink band of GSM900 are selected as the data of this experiment for the training and prediction of the model.

### 4.1.1. Data Preprocessing

Before analyzing the spectrum data, it is necessary to preprocess the existing original signal spectrum. There are only two states in the actual spectrum: idle state and occupied state. Usually, a threshold value is set first, and the spectrum states are distinguished by comparing the spectrum value with the threshold value. That is, if the spectrum value of the channel is lower than the threshold at a certain time, the channel is considered to be unused and idle and is represented by "0", while the opposite is represented by "1".

In the research, the determination of the threshold value plays a great role in the experimental results, because when the threshold value is set too large, some useful signals will be missed, but when the threshold value is set too small, noise signals will be mistaken for useful signals. How to select the threshold value is a key step in data processing and analysis. Because there is a large amount of original spectrum data, select some data to display, as shown in Table 1.

**Table 1.** Partial spectrum data.

| Channel \ Slot | 1 | 2 | 3 | 4 | 5 | 6 | 7 | 8 | 9 |
|---|---|---|---|---|---|---|---|---|---|
| 1 | $-97.799$ | $-96.879$ | $-98.058$ | $-101.042$ | $-100.307$ | $-100.696$ | $-100.751$ | $-101.346$ | $-100.867$ |
| 2 | $-101.126$ | $-100.244$ | $-100.150$ | $-101.270$ | $-103.100$ | $-102.444$ | $-102.598$ | $-102.805$ | $-102.943$ |
| 3 | $-99.644$ | $-100.088$ | $-99.616$ | $-100.602$ | $-100.106$ | $-100.632$ | $-101.950$ | $-101.225$ | $-100.784$ |
| 4 | $-99.763$ | $-97.423$ | $-99.049$ | $-98.909$ | $-97.472$ | $-98.264$ | $-100.034$ | $-97.922$ | $-99.077$ |
| 5 | $-99.010$ | $-96.495$ | $-99.228$ | $-99.385$ | $-99.082$ | $-98.963$ | $-100.305$ | $-98.338$ | $-98.964$ |
| 6 | $-96.176$ | $-97.853$ | $-97.399$ | $-99.4540$ | $-99.284$ | $-98.286$ | $-97.975$ | $-99.013$ | $-100.780$ |

In the data analysis of this experiment, the threshold value is set as $-100$ dB by referring to the method of researchers' threshold value, and the channel state is distinguished by comparing the spectrum value of $-100$ dB with the original useful signal. When $-100$ dB is less than the spectrum value of the signal, it is represented by "1"; when it is greater than the spectrum value, it is represented by "0" to realize the conversion of the spectrum value into the binary representation of 0 and 1, where "0" means that the channel is idle at this moment, and "1" means that the channel is used by PU. So, Table 1 can be converted to Table 2 to indicate the occupied or idle state of the channel.

**Table 2.** Partially processed data.

| Channel \ Slot | 1 | 2 | 3 | 4 | 5 | 6 | 7 | 8 | 9 |
|---|---|---|---|---|---|---|---|---|---|
| 1 | 1 | 1 | 1 | 0 | 0 | 0 | 0 | 0 | 0 |
| 2 | 0 | 0 | 0 | 0 | 0 | 0 | 0 | 0 | 0 |
| 3 | 1 | 0 | 1 | 0 | 0 | 0 | 0 | 0 | 0 |
| 4 | 1 | 1 | 1 | 1 | 1 | 1 | 0 | 1 | 1 |
| 5 | 1 | 1 | 1 | 1 | 1 | 1 | 0 | 1 | 1 |
| 6 | 1 | 1 | 1 | 1 | 1 | 1 | 1 | 1 | 0 |

### 4.1.2. Calculate the Channel Occupancy

According to the processed experimental data, we can calculate the corresponding occupancy of each channel. Here, we define channel occupancy as the number of occupied channels divided by the total number of idle plus occupied channels in the entire channel. Figure 11 shows the channel occupancy in the upstream and downstream bands of the GSM900. The abscissa is the channel number, and the ordinate is the channel occupancy.

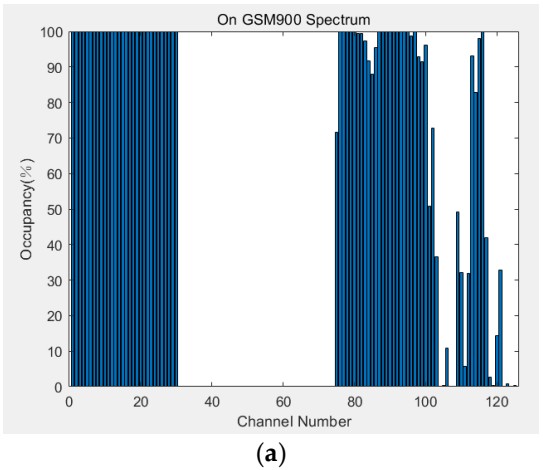
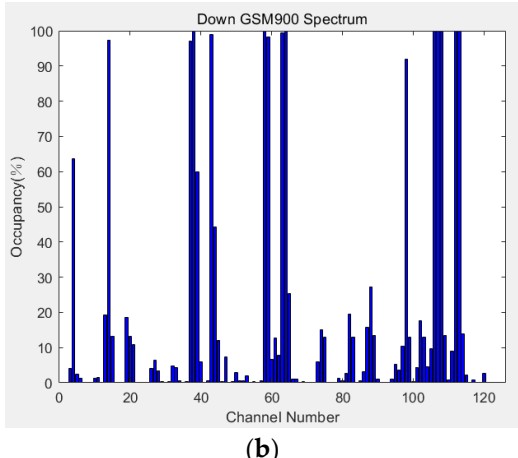

(**a**)                                                                                                   (**b**)

**Figure 11.** Occupancy of the channel: (**a**) the channel occupancy in the uplink band of GSM900; (**b**) the channel occupancy in the downlink band of GSM900.

It can be seen that the occupancy of each channel is not high, and even the occupancy of some channels is 0. The results show that the corresponding spectrum utilization is not very ideal, and there is a lot of space to improve the utilization. In order to calculate the occupancy of each channel more intuitively, we conducted further statistical analysis on the GSM900 uplink band and GSM900 downlink band, respectively. The proportion of the number of channels with occupancy less than *O* in each band to the total number of channels in this band is shown in Table 3, where $O \in [10\%, 100\%]$.

**Table 3.** Occupancy of the GSM900 channel.

| Occupancy<br>Frequency Band | 10% | 20% | 30% | 40% | 50% | 60% | 70% | 80% | 90% | 100% |
|---|---|---|---|---|---|---|---|---|---|---|
| Uplink band of GSM900 | 0.44 | 0.456 | 0.456 | 0.488 | 0.504 | 0.512 | 0.512 | 0.528 | 0.544 | 1 |
| Downlink band of GSM900 | 0.696 | 0.848 | 0.864 | 0.864 | 0.872 | 0.88 | 0.888 | 0.888 | 0.888 | 1 |

As can be seen from the data in Table 3, the channel occupancy in the downlink band of GSM900 is not ideal, with nearly 85% of the channels occupying less than 20%, and nearly half of the channels in the uplink band occupying less than 40%. There are a large number of unused or underused channels in the GSM900 band, which wastes spectrum resources to a great extent, but also has more space to use. Therefore, we can start from the low-occupied frequency band and improve the channel occupancy rate and spectrum utilization by spectrum prediction technology.

4.1.3. Evaluation Criterion

In order to make the network model designed in this paper able to evaluate the prediction more realistically, the following evaluation indicators are defined:

Root Mean Squared Error (RMSE) is the square root of the ratio between the sum of the squared differences between the predicted value ($x_i$) and the true value ($y_i$) and the number of predictions (*n*), which is calculated as follows.

$$RMSE = \sqrt{\frac{1}{n}\sum_{i=1}^{n}(x_i - y_i)^2} \qquad (17)$$

It is a common metric to evaluate the performance of response prediction. The lower the RMSE, the smaller the error between the predicted value and the true value of the model, and the better the model is.

Mean Absolute Error (MAE) is the ratio between the sum of the absolute errors of the predicted value and the actual value and the number of predictions. The specific formula is as follows. MAE can better reflect the actual situation of the predicted value error.

$$MAE = \frac{1}{n}\sum_{i=1}^{n}|x_i - y_i| \tag{18}$$

The prediction accuracy is the ratio of the number of successful predictions to the total number of predictions, which gives a very intuitive indication of the accuracy and performance of the model prediction, and is denoted by "Acc".

$$Acc = \frac{Predict\ the\ right\ amount}{Total\ number\ of\ predictions} \tag{19}$$

4.1.4. Comparison of Simulation Results

We used the processed data in Section 4.1.1 for experiments to study the relationship between the total sequence length and the model prediction rate. In this experiment, by changing the length of the sequence, the prediction effects of the four models under different sequence lengths are obtained, and the performance of the model prediction is judged by comparing the prediction accuracy of the spectrum signal prediction of these models. A detailed simulation of the prediction rate is shown in Figure 12.

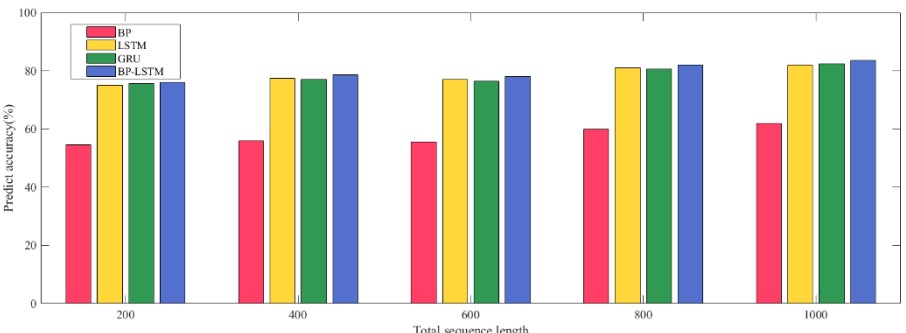

**Figure 12.** Total sequence length vs. prediction rate.

As can be seen from Figure 13, when the sequence length changes, the training results of different models are different, resulting in different prediction effects. In general, the B-LTF model has the best prediction effect. It can be seen that when the sequence length is 1000, the prediction accuracy is more than 80%, and the prediction effect of the LSTM model and GRU model is not much different. The BP model is lower, but the prediction rate is also about 60%. At the same time, for the same model, the prediction accuracy shows an upward trend with the increase in the total sequence length. This is due to the increase in historical information, the model can better learn and find the relationship between each point, thereby improving the prediction accuracy. However, it can be found that the growth trend will gradually become smaller, so whether the final prediction accuracy will saturate when the sequence is long enough needs to be tested with more data sets. In order to analyze and compare the prediction effect of each model more intuitively, the predicted values of the four models are compared with the true values in Figure 13. In the four simulation diagrams, the horizontal coordinate is the length of the sequence, the vertical coordinate is the channel state value, the blue line represents the current real channel state, and the red line represents the channel state predicted by the four models. The more the red overlaps with blue, the higher the prediction accuracy of the model.

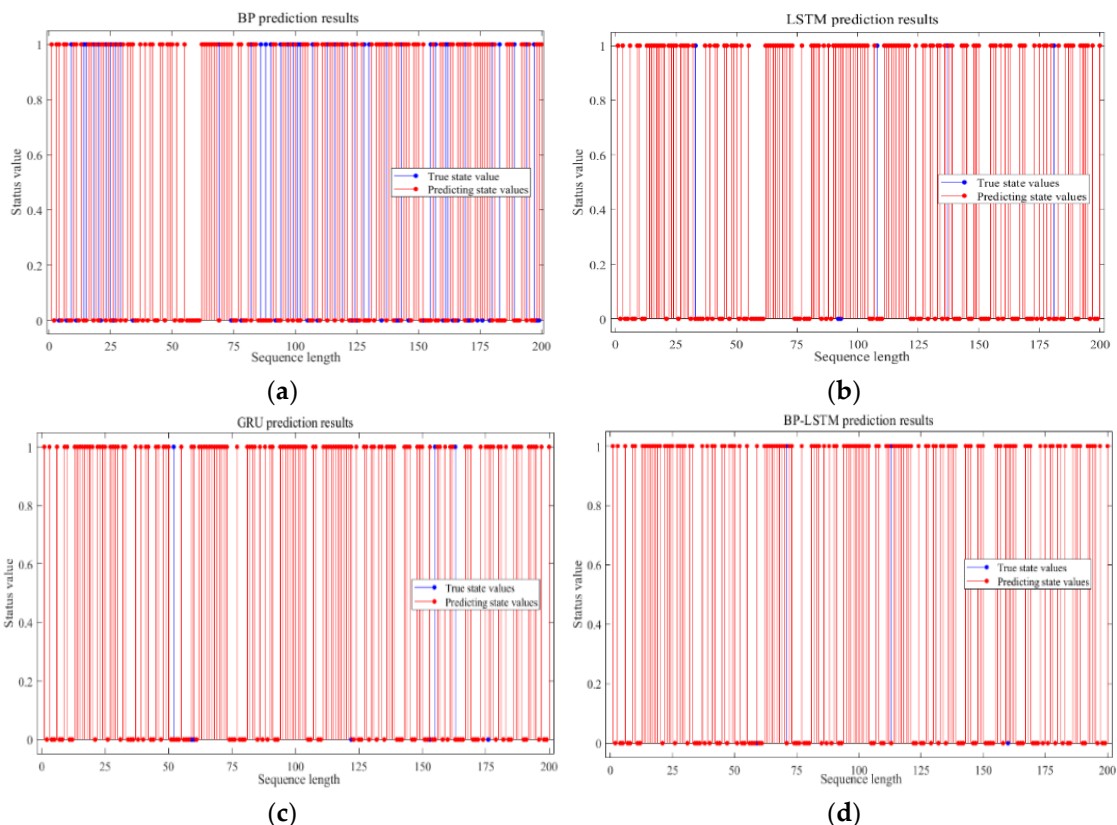

**Figure 13.** Plot of predicted vs. true values for the four models: (**a**) BP prediction results; (**b**) LSTM prediction results; (**c**) GRU prediction results; (**d**) B-LTF prediction results.

In addition, for the LSTM model, GRU model and B-LTF model, the size of the sliding window is changed to compare the prediction effects of the three models, as shown in Figure 14.

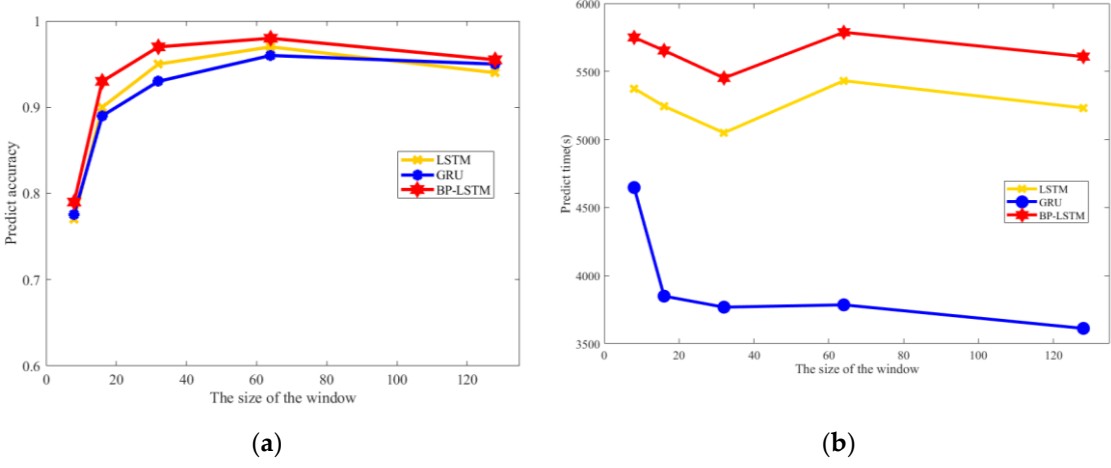

**Figure 14.** Window size variation: (**a**) the prediction accuracy of the three models at different window sizes; (**b**) the corresponding training times.

It can be seen from (a) that with the increase in window size, the prediction accuracy of the three network models shows an upward trend, which is more obvious from 8 to 16, and the upward trend is moderated after 16. The prediction accuracy of the three network models is excellent, and the performance of B-LTF is better. When the window size is 64, the prediction rate reaches about 98%, which is about 1% higher than that of LSTM and

GRU models, and the performance of LSTM and GRU is similar. In addition, it can also be seen that when the window size is changed again, the prediction accuracy is saturated, no longer changes, and may even decrease a little, which is caused by the structure of the LSTM network model itself. From (b), it can be seen that the overall training is often relatively long. GRU, as the "brother" of LSTM, in the case of the same complexity, because of its simpler structure, the operation speed is also faster. B-LTF, as a combination model, takes the highest operation time among the three models. When the window size is above 32 and the number of training rounds reaches about 120, the value of loss almost no longer changes and always maintains a value. Therefore, in actual prediction, the number of training rounds can be appropriately reduced to reduce the operation time of the model.

*4.2. Prediction of the Trend of the Spectrum Signal*

For the spectrum signal, after processing according to the threshold value, we can know the channel usage of the current time slot, so as to predict the future state. In addition, it can also directly predict the size and trend of the future spectrum signal through the internal relationship between the current spectrum value of the spectrum signal. The channel state only predicts whether the channel is occupied, while signal spectrum prediction goes deeper and can predict the signal amplitude. The purpose of the two is not the same. If it is accessible, that is, as long as the idle channel is found, and the signal spectrum prediction can be used to complete the signal spectrum, that is, to complete the signal spectrum with a small amount of sparse data and prediction. Since the spectrum signal can be regarded as a time series, the neural network can also be used for sequence-to-sequence prediction.

4.2.1. Data Processing and Parameter Setting

The data of this simulation experiment is also used in RWTH Aachen University in Germany mentioned in Section 4.1 because the power amplitude trend of the future spectrum is predicted, so there is no need for excessive data preprocessing.

As for the data set, it is first divided into the training set and test set. Here, the first 80% is still selected as the training set, and the last 20% is used as the prediction set. The time series is predicted by the network model designed in Sections 2.2 and 3.2. Therefore, instead of comparing the prediction accuracy, the RMSE and MAE of these models for spectrum signal prediction are compared.

4.2.2. Experimental Result

Figure 15 shows the comparison between the predicted value of the spectrum and the real value of the spectrum of the above four models. It can be seen intuitively that the predicted value of the B-LTF model is closer to the trend of the real value. In addition, under the same spectrum data and the same number of hidden layers, the RMSE and MAE values of the four models are shown in Table 4.

From the prediction results and corresponding values of the above four models, it can be seen that the prediction results of the BP model are too jumping and not very stable, resulting in a small MAE value but a large RMSE value. However, the prediction results of the other three models are relatively stable, although there are errors with the true value, but the difference changes little. It also proves that the prediction performance of the BP network model for spectrum trend still needs to be improved, compared with the trend prediction of the other three models are better. Among them, the B−LTF network model has the best prediction effect, and the error between the prediction result and the true value is the smallest, which also verifies that its evaluation index is better than the other three models.

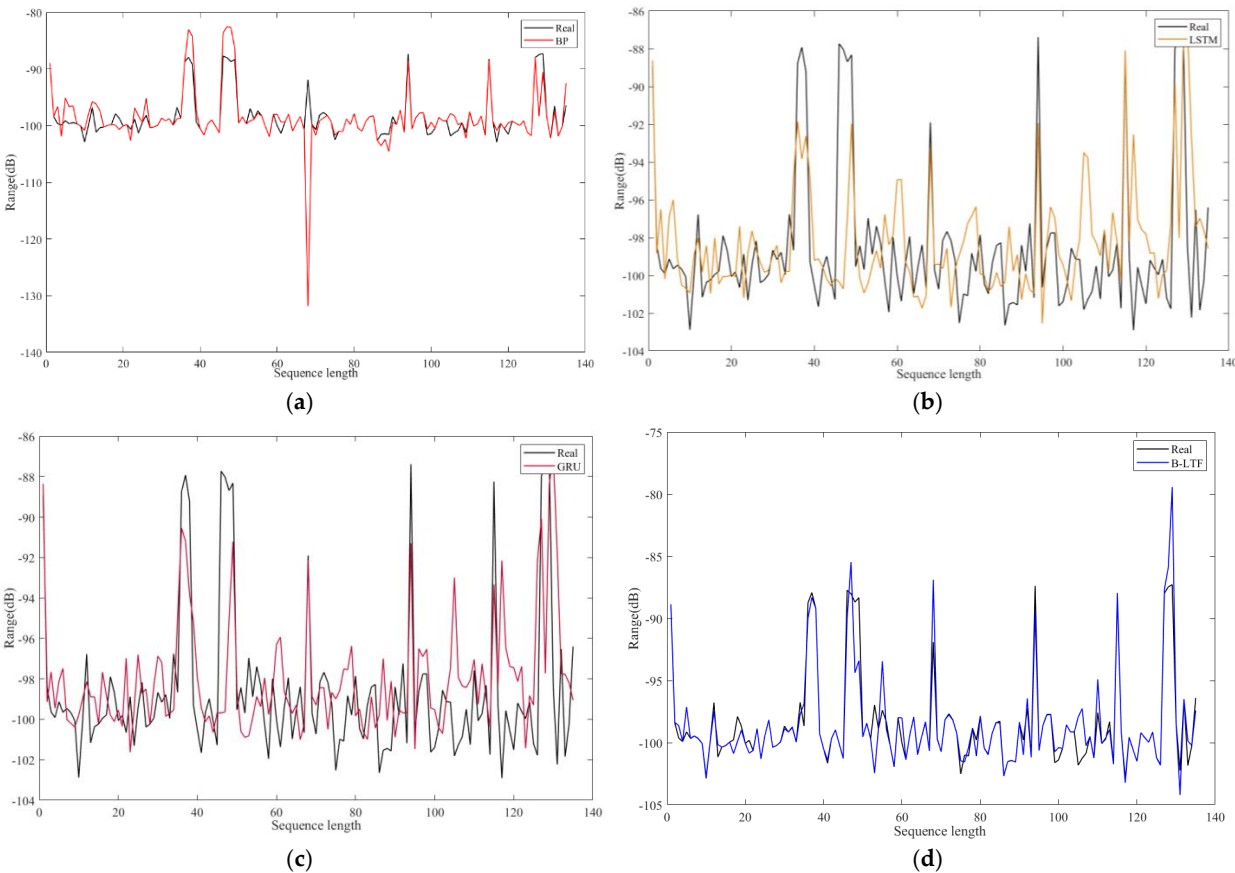

**Figure 15.** Plot of actual vs. predicted values: (**a**) true value vs. BP model prediction results; (**b**) true value vs. LSTM model prediction results; (**c**) true value vs. GRU model prediction results; (**d**) true value vs. B−LTF model prediction results.

**Table 4.** Comparison of RMSE and MAE values of the four models.

| Model \ Evaluation Index | RMSE | MAE |
|---|---|---|
| BP | 3.927 | 1.6417 |
| LSTM | 3.2344 | 2.1965 |
| GRU | 3.3741 | 2.3031 |
| B-LTF | 1.6746 | 0.6309 |

For the prediction of channel state, the influence of different sequence lengths and window sizes on the prediction performance is studied. The results show that with the increase in sequence length and window size, the prediction performance of the four models is on the rise. In addition, according to the analysis of the above four models, the LSTM model and its improved model have better prediction performance than the BP neural network. The B-LTF has better prediction performance but is replaced by a longer training time. For the trend prediction of spectrum prediction, the experimental results show that the improved recurrent neural network has certain advantages in processing time series, and the prediction effects of LSTM and GRU models are good. The overall prediction effect of the B-LTF model is closer to the true value. There will be some big jumps in the spectrum values, making the final prediction not very close to the true value. In general, although the prediction performance of these models is ideal, and the B-LTF model improves the prediction performance of the original LSTM model again, there is a long training process, and it can be found that the increase in sequence length will reduce the prediction performance of the above models.

## 5. Conclusions

In this paper, through the processing and analysis of an actual spectrum observation data set, we find that the spectrum utilization rate of the GSM900 band is not high, and there is a large waste of resources, which proves that the spectrum in today's world has a major problem. Based on this, we propose the B-LTF algorithm to predict the spectrum using neural networks. In it, we analyze the performance of four network models, namely the B-LTF model, GRU model, LSTM model and BP model, in predicting the channel occupancy status and predicting the future spectrum trend in detail. From the overall simulation results, the prediction effect of the long short-term memory network and its improved model is better than that of the BP network model, which also proves that they have a strong ability to deal with time series. When predicting the channel occupancy state, the B-LTF model improves the prediction accuracy by about 1% to 2% compared with the LSTM and GRU models. In the spectrum signal trend prediction, the B-LTF model can predict the overall trend and amplitude change of the future spectrum signal more effectively, but the sequence length has a great influence on the prediction results in the overall simulation and the operation time is too long. Therefore, how to reduce the operation time and reduce the impact of sequence length on the prediction results while improving the spectrum prediction performance are the main issues worth exploring in future research.

**Author Contributions:** Q.C. and X.W. put forward the idea of this paper. Q.C. finished the design of the study and the algorithms. Q.C. and X.W. contributed to the experimental work and the data analysis. Q.C. and X.Y. made figures and tables. Q.C. and X.W. drafted the manuscript. All authors have read and agreed to the published version of the manuscript.

**Funding:** This research was funded by Heilongjiang Postdoctoral Foundation, grant number LBH-Z19168 and Heilongjiang Provincial Natural Science Foundation of China, grant number YQ2022F014.

**Institutional Review Board Statement:** Not applicable.

**Informed Consent Statement:** Not applicable.

**Data Availability Statement:** Not applicable.

**Conflicts of Interest:** The authors declare no conflict of interest.

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
