# Peer review of "Research on Spectrum Prediction Technology Based on B-LTF"

_electronics, doi:10.3390/electronics12010247_

Round 1
Reviewer 1 Report
There are some mistakes in this paper. Abbreviations are not properly defined. References and not up to date.
By looking at paper overall structure, presentation and above all the provided contents, I would say the authors of the paper requires minor changes to accept it for publication.
1. In abstract section, the authors used GRU abbreviation but does not explain it.
2. In section 2.2.1, the authors defined prediction steps. It would be BP-neural
network algorithm, because steps will not be clear to the viewers.
3. All the equations in this paper are not defined. It must be elaborate the function of each mathematical equation.
4. The explanation of B-LTF algorithm is missing in this research paper.
5. In the abstract section, the authors defined BP-LTF but in section 3.2, a number of times B-LTF is defined, differentiate it.
6. In table 4, the authors used an abbreviation of RMSE and MAE but don’t explain these abbreviations anywhere.
7. Most of the references are too old. It must be updated before the publication.
Author Response
Comments by Reviewer 1_____________________________________________________
The authors would like to thank Reviewer 1 for her/his constructive comments, which have helped us further improve the presentation of the paper. A point-by-point response is provided below, with the changes highlighted in yellow in the revised manuscript.
Comments to the AuthorThere are some mistakes in this paper. Abbreviations are not properly defined. References and not up to date. By looking at paper overall structure, presentation and above all the provided contents, I would say the authors of the paper requires minor changes to accept it for publication..
Comment 1: In abstract section, the authors used GRU abbreviation but does not explain it.
Response 1: Thanks for your valuable comments. In light of these comments, some specifics have been added in Abstract. The detailed changes are as follow:Abstract, Page 1, Paragraph 1. This paper proposes a spectrum predicting method called Back Propagation-Long short-term memory Time Forecasting (B-LTF) by using Back Propagation-Long Short-term Memory (BP-LSTM) network model. According to the historical spectrum data, the future spectrum trend and the channel state of the future time node are predicted. The purpose of our research is to achieve dynamic spectrum access by improving the accuracy of spectrum prediction and better assisting cognitive radio technology. By comparing with BP, LSTM and Gate Recurrent Unit (GRU) network models, we clarify that the improved model of recurrent time network can deal with time series more effectively.
Comment 2: In section 2.2.1, the authors defined prediction steps. It would be BP-neural network algorithm, because steps will not be clear to the viewers.
Response 2: Thanks for pointing out these suggestions! We looked at the corresponding paragraphs in detail and turned the prediction step into the algorithmic process of BP network.
Section 2.2.1, Page 6. Comment 3: All the equations in this paper are not defined. It must be elaborate the function of each mathematical equation.Response 3: Thank you for pointing this out and we added some description of equations.Section 2.2.3“Where σ is the sigmoid function, tanh is the hyperbolic tangent activation function, and W is the weight matrix of the corresponding module. Equation (2) shows that when time step is t, the input vector is the result of linear transformation obtained by multiplying xt and the state ht-1 saved at the previous time step t-1 with the weight matrix respectively. The reset gate uses an activation function to convert the sum of two results into [0,1]. Equation (3) is the relevant formula of the update gate, which is similar to the expression of the reset gate. Equation (4) is expressed as the memory content in the current cell, and the current information and historical information are added and transformed by tanh function. Equation (5) represents the final output of the memory unit.”Section 3.1“Where xt is the input vector at time step t, the hidden layer state vector is ht, ct is the memory cell, σ is the sigmoid function, tanh is the hyperbolic tangent activation function, W and U are the weight matrix of the corresponding module, and b is the bias vector of the corresponding template. Equations (11) and (14) are related calculations for input gates, Equation (15) is the Cell state equation at time ‘t’, Equation (12) is related calculations for forget gate, Equations (13) and (16) are related calculations for output gate.”
Comment 4: The explanation of B-LTF algorithm is missing in this research paper.
Response 4: Thank you for your valid suggestion! We would add the explanation of B-LTF algorithm in the revised manuscript. Section 1, Page3.“Despite the high performance of LSTM, it may bring challenges in terms of computational burden and handling missing data, and since spectral data is equivalent to long time series, and LSTM has reduced prediction accuracy for long sequences, so we propose B-LTF algorithm to improve the problem of long sequence prediction. It slows down the impact of sequence length change on the prediction performance of the network, so that the overall spectrum prediction rate is improved.”Section 3.2, Page10.“This paper mainly focuses on the prediction and research of spectrum signals. Since spectrum signals are equivalent to a long time series, its prediction is a series-to-sequence prediction. Therefore, according to this situation, we proposed B-LTF algorithm, which is mainly a prediction model composed of BP neural network and LSTM network. Among them, BP neural network is not only simple in structure, but also has strong nonlinear mapping ability. However, it does not consider the time correlation between the data, which often has a big problem for the prediction of time series. When facing the same data set, it will still wait for the same result after transformation. In addition, some traditional neural networks are prone to problems such as the disappearance of gradient in the prediction, but LSTM network changes this situation. It can solve the gradient problem in the prediction, and because of the existence of gates, LSTM neural network has better performance in dealing with time series problems. Therefore, combining the advantages of BP neural network and LSTM neural network, the construction of BP-LSTM combined network prediction model can not only solve the deficiencies of BP model, but also improve the convergence speed of the overall model, so as to improve the prediction rate of the model.”
Comment 5: In the abstract section, the authors defined BP-LTF but in section 3.2, a number of times B-LTF is defined, differentiate it.
Response 5: Thank you for your valid suggestion! We defined B-LTF in the Abstract, however in the Section 3, We gave the definitions of the two, one is the name of the method and the other is the network model. Please refer to which have been corrected in the revised manuscript for details.
“This paper proposes a spectrum predicting method called Back Propagation-Long short-term memory Time Forecasting (B-LTF) by using Back Propagation-Long Short-term Memory (BP-LSTM) network model.”
Comment 6: In table 4, the authors used an abbreviation of RMSE and MAE but don’t explain these abbreviations anywhere.
Response 6: The authors would like to thank the reviewer for her/his insight comment. However, I gave the full name of RMSE and MAE in section 4.1.3 of this article.
“Root Mean Squared Error (RMSE) is the square root of the ratio between the sum of the squared differences between the predicted value (xi) and the true value (yi) and the number of predictions (n), which is calculated as follows.”“Mean Absolute Error (MAE) is the ratio between the sum of the absolute errors of the predicted value and the actual value and the number of predictions. The specific formula is as follows. MAE can better reflect the actual situation of the predicted value error.”
Comment 7: Most of the references are too old. It must be updated before the publication.
Response 7: Thank you for your valid suggestion! We would update the references in the revised manuscript.
“Woods, T.; Furman, S. All-domain Spectrum Command and Control via Hierarchical Dynamic Spectrum Sharing with Implemented Dynamic Spectrum Access Toolchain. MILCOM 2021 - 2021 IEEE Military Communications Conference (MILCOM), San Diego, CA, USA, 29 November 2021 - 02 December 2021.Madhavan, A.; Govindarajan, Y. Wideband Spectrum Sensing in Dynamic Spectrum Access Sys-tems Using Bayesian Learning. Journal of Physics: Conference Series 2021, 1964, 62067-62075.Toma, O. H.; Lopez-Benitez, M. Cooperative Spectrum Sensing: A New Approach for Minimum Interference and Maximum Utilisation. 2021 IEEE International Conference on Communications Workshops (ICC Workshops), Montreal, QC, Canada, 14-23 June 2021.Solanki, S.; Dehalwar, V. Spectrum Sensing in Cognitive Radio Using CNN-RNN and Transfer Learning. IEEE Access 2022, 10, 113482 - 113492.Yang, C.; Peng, T.; Zuo, P. A Spectrum Prediction Method for Bursty Frequency Bands. 2021 IEEE Wireless Communications and Networking Conference Workshops (WCNCW), Nanjing, China, 29-29 March 2021. Gao, Y. L.; Zhao, C. Y.; Fu, N. Joint multi-channel multi-step spectrum prediction algorithm. 2021 IEEE 94th Vehicular Technology Conference (VTC2021-Fall), Norman, OK, USA, 27-30 September 2021.Huang, Q. Y. Research On Cognitive Radio Spectrum Prediction Based On LSTM Neural Network. Master of Engineering, Harbin Institute of Technology, Harbin, China, June, 2020.Radhakrishnan, N.; Kandeepan, S. Performance Analysis of Long Short-Term Memory-Based Markovian Spectrum Prediction. IEEE Access 2021, 9, 149582 – 149595.Aygul, M. A.; Nazzal, M. Efficient Spectrum Occupancy Prediction Exploiting Multidimensional Correlations through Composite 2D-LSTM Models. Sensors 2021, 21, 1-18.Radhakrishnan, N.; Kandeepan, S. An Improved Initialization Method for Fast Learning in Long Short-Term Memory-Based Markovian Spectrum Prediction. IEEE Transactions on Cognitive Com-munications and Networking 2021, 7, 729-738.Radhakrishnan, N.; Kandeepan, S.; Yu, X. H. Soft Fusion based Cooperative Spectrum Prediction using LSTM. 2021 15th International Conference on Signal Processing and Communication Systems (ICSPCS), Sydney, Australia, 13-15 December 2021.”

Reviewer 2 Report
This paper proposes a spectrum-predicting method, B-LTF, by using BP-LSTM network model. The authors claim that they can achieve dynamic spectrum access by improving the accuracy of spectrum prediction and better assisting cognitive radio technology. The simulation results seem to support their sayings. This paper is well-structured and interesting to the reader. I have some comments.
(1) In section one, although the authors spend many paragraphs describing the background of this research topic, point out the importance of applying deep learning to spectrum prediction technology, and present some works related to this topic, the research motivation is unclear. The authors have to point out the shortcomings of existing works and present the reasons why combining the BP network with the LSTM can improve the accuracy of spectrum prediction effectively.
(2) In B-LTF algorithm on page 10, line 2 states that the output is the Trained BP-LSTM model, but the bottom of this algorithm says that the output is the prediction results. Is it correct?
(3) In figure 10, the "total onput" should be replaced by "total output".
(4) In the simulation, the authors compare the prediction effects of the four models under different sequence lengths. I suggest that except for the parameter setting of the proposed approach, the parameter setting of other models also needs to be present.
Author Response
Comments by Reviewer 2______________________________________________________
The authors would like to thank Reviewer 2 for her/his constructive comments, which have helped us further improve the presentation of the paper. A point-by-point response is provided below, with the changed highlighted in yellow in the revised manuscript.
Comments to the AuthorThis paper proposes a spectrum-predicting method, B-LTF, by using BP-LSTM network model. The authors claim that they can achieve dynamic spectrum access by improving the accuracy of spectrum prediction and better assisting cognitive radio technology. The simulation results seem to support their sayings. This paper is well-structured and interesting to the reader. I have some comments.
Comment 1: In section one, although the authors spend many paragraphs describing the background of this research topic, point out the importance of applying deep learning to spectrum prediction technology, and present some works related to this topic, the research motivation is unclear. The authors have to point out the shortcomings of existing works and present the reasons why combining the BP network with the LSTM can improve the accuracy of spectrum prediction effectively.
Response 1: The authors would like to thank the reviewer for her/his insight comment. To address these comments, some descriptions have been added to Section 1 as follows. Section 1, Page 2.“However, in the process of spectrum sensing, in order to obtain more spectrum resources, the range of spectrum sensing is often increased, but the corresponding perception time will be extended, and a lot of calculation and more and more complex algorithm process will be generated. In order to solve such problems, Haykin first proposed spectrum prediction technology based on spectrum sensing technology in 2005, and then around 2012, researchers strengthened the concept of spectrum prediction technology. First of all, spectrum prediction technology can find the relationship between the spectrum through the study of the historical spectrum segment, predict the future spectrum state, find a range of possible free frequency band, and then carry out spectrum perception of this range, can directly find the free spectrum resources in a short time, so as to solve the problem of high spectrum perception consumption. Secondly, when the received signal is incomplete, we can also predict the amplitude and trend of the complete signal according to the spectrum prediction technology, and carry out an information demodulation. Finally, it allows users to access the underutilized spectrum opportunistically in the time domain or spectrum domain without significantly affecting other users, which provides the possibility for subsequent spectrum decision making, spectrum migration and spectrum sharing, thus assisting cognitive radio technology.”Section 1, Page 3-4. “In addition, for different industrial frequency bands and different communication systems, researchers have also proposed other prediction algorithms based on LSTM networks, which play a great contribution to the research of spectrum prediction technology. Despite the high performance of LSTM, it may bring challenges in terms of computational burden and handling missing data, and since spectral data is equivalent to long time series, and LSTM has reduced prediction accuracy for long sequences, so we propose B-LTF algorithm to improve the problem of long sequence prediction. It slows down the impact of sequence length change on the prediction performance of the network, so that the overall spectrum prediction rate is improved. Secondly, in this paper, we not only predict the channel state, but also predict the spectrum signal. The prediction of channel state is first processed by gate threshold and then predicted, so as to know the channel state of the node in the future time, and the signal spectrum pre-diction goes deeper into the deeper level, that is, the trend and amplitude change of the future signal can be known through the sparse spectrum signal data, which is of great help to the subsequent cognitive radio technology.”
Comment 2: In B-LTF algorithm on page 10, line 2 states that the output is the Trained BP-LSTM model, but the bottom of this algorithm says that the output is the prediction results. Is it correct?
Response 2: Thanks for pointing out these errors. We changed the output to the prediction results.
Comment 3: In figure 10, the "total onput" should be replaced by "total output".
Response 3: Thanks for pointing out these errors, which have been corrected in the revised manuscript.
Comment 4: In the simulation, the authors compare the prediction effects of the four models under different sequence lengths. I suggest that except for the parameter setting of the proposed approach, the parameter setting of other models also needs to be present.
Response 4: Thank you for your valid suggestion! In section 2, we would add the parameter setting of other models, as follows:Section 2.2.2, Page 6. “The LSTM network model designed in this paper will consist of five layers, first an input layer, then an LSTM layer, followed by a forgetting layer, immediately followed by a fully connected layer, and finally an output layer. The number of nodes in the LSTM layer is set to 1024, the activation function of the hidden layer uses tanh activation function, the initialization uses orthogonal initialization, the activation function of the full connection layer is set to the ’softmax’ activation function, and the forgetting rate is set to 0.2 in the forgetting layer. The optimizer of the whole LSTM network uses Adaptive momentum (Adams) algorithm optimizer, and the number of training sessions is set to 200-300.” Section 2.2.3, Page 7.“The GRU network is also similar to the LSTM network model in model design. Firstly, a GRU layer is set, the number of nodes in the network layer is set to 1024, the tanh function is also used as the activation function, the optimizer chooses Adams, the number of training sessions is set to 200-300, and a fully connected layer follows the GRU layer. There is also an input layer and an output layer, and the complete GRU network model is shown in Figure 9.

Reviewer 3 Report
The article is interesting but needs several changes.
1. The introduction needs to be rewritten. In the introduction, a series of recent articles related to the topic addressed are not cited.
2. The paragraph "System model", must be narrowed. The models are known and can be found in several articles.
3. Combined BP-LSTM neural network is not new. See the following articles:
H. Jia and X. Zhou, "Water Quality Prediction Method Based on LSTM-BP," 2020 12th International Conference on Intelligent Human-Machine Systems and Cybernetics (IHMSC), 2020, pp. 27-30, doi: 10.1109/IHMSC49165.2020.00014.
Hua C, Zhu E, Kuang L, Pi D. Short-term power prediction of photovoltaic power station based on long short-term memory-back-propagation. International Journal of Distributed Sensor Networks. 2019;15(10), doi:10.1177/1550147719883134
4. Considering the above, the claimed contributions must be rewritten. I recommend you rewrite paragraph 3 of the article.
5. In paragraph 4 of the article, please present more details about the experimental part.
6. I recommend that the experimental results be presented in a single figure. Predictions (a)-(e) should be presented in a single figure (use different colors).
7. The conclusions must be much more relevant.
In conclusion, the article in its current form requires a lot of changes. As such, I propose rewriting the entire article and sending it for publication at another time.
Author Response
Comments by Reviewer 3_____________________________________________________
The authors would like to thank Reviewer 1 for her/his constructive comments, which have helped us further improve the presentation of the paper. A point-by-point response is provided below, with the changes highlighted in yellow in the revised manuscript.
Comments to the Author
The article is interesting but needs several changes.
Comment 1: The introduction needs to be rewritten. In the introduction, a series of recent articles related to the topic addressed are not cited.
Response 1: Thanks for pointing out these errors. We updated and increased some related to the topics in the introduction. Here's some of it:“In order to solve the long-term predictions of spectrum data, [26] proposes a convolutional LSTM model for spectrum prediction—ConvLSTM. In the 450-520MHZ frequency band, the long-term spatial-spectral-time joint prediction of the spectrum signal is carried out. The results show that the model exhibits a stable value of RMSE for different signals, but it will rise with the increase of time step. In [27], STS-PredNet was proposed by combining ConvLSTM and predictive recurrent neural network (PredRNN). This model shows stable Root Mean Squared Error (RMSE) performance in multi-time step spatial-temporal spectrum prediction, and effectively improves the prediction performance in different prediction ranges. For the ISM band with high burst characteristics, [28] proposed an algorithm named Classified-based deep reinforcement learning (C-DRL) to quickly and efficiently complete the matching between the state and the prediction model, achieving the effect of fast convergence, simple operation and high prediction rate. In order to reduce the time cost of convolutional network prediction in multi-channel, Gao et al. use LSTM, Seq-to-Seq modelling, exploit the intrinsic correlation and time correlation between channels, and propose a multi-channel multi-step joint spectrum prediction algorithm, which can effectively improve the prediction accuracy of multi-channel [29].”
Comment 2: The paragraph "System model", must be narrowed. The models are known and can be found in several articles.
Response 2: Thanks for pointing out these suggestions. In the Section 2, We deleted the principle of the RNN model and part of the LSTM model, and the rest is the parameter setting of the subsequent experimental model, and we also added the parameter setting of the model. Here's some of it:Section 2.2.3, Page 7.“The GRU network is also similar to the LSTM network model in model design. Firstly, a GRU layer is set, the number of nodes in the network layer is set to 1024, the tanh function is also used as the activation function, the optimizer chooses Adams, the number of training sessions is set to 200-300, and a fully connected layer follows the GRU layer. There is also an input layer and an output layer, and the complete GRU network model is shown in Figure 9.”
Comment 3: Combined BP-LSTM neural network is not new. See the following articles:
H. Jia and X. Zhou, "Water Quality Prediction Method Based on LSTM-BP," 2020 12th International Conference on Intelligent Human-Machine Systems and Cybernetics (IHMSC), 2020, pp. 27-30, doi: 10.1109/IHMSC49165.2020.00014.
Hua C, Zhu E, Kuang L, Pi D. Short-term power prediction of photovoltaic power station based on long short-term memory-back-propagation. International Journal of Distributed Sensor Networks. 2019;15(10), doi:10.1177/1550147719883134.
Response 3: Thank you for your questions and related literature!After reading the two papers you provided us in detail, we can conclude as follows: These two papers respectively use the LSTM-BP network model to predict the water quality and the accurate prediction of the power generation capacity of photovoltaic power generation system, both of which basically explain their own network has better advantages by comparing with BP and LSTM. In our paper, we proposed B-LTF algorithm, which is also the combination network of BP and LSTM, but the algorithm we proposed is applied to spectrum prediction technology. Since a large number of spectrum signals are many long time series, which is equivalent to a series-to-series prediction, the traditional neural network is not particularly ideal for the processing of long time series. Therefore, we propose this algorithm mainly to solve the prediction of long time series of spectrum signals, and explore the influence relationship between the length of spectrum series and the frequency prediction rate. In this paper, we not only discuss the B-LTF algorithm, but also illustrate the advantages of LSTM in processing time series by comparing BP and LSTM. Meanwhile, we also compare LSTM and GRU model, emphasizing that the prediction performance of the two is similar, but because of the different structure, the training time of GRU is faster. Finally, by comparing the prediction rates of B-LTF under different sequence lengths, it can be clearly found that the proposed algorithm has a great improvement in spectrum prediction. This provides a more scientific method for sensing spectrum holes, which can be applied in spectrum prediction technology, and provides a basis and reference for realizing spectrum sharing. In addition, we also discussed the spectrum occupancy characteristics according to the real spectrum data, determined the real state of the channel under each time node of the real data by the gate threshold method, analyzed the occupancy of each channel in the GSM900 frequency band in detail by statistical classification, and summarized the spectrum utilization rate of the whole frequency band, supporting the lack of spectrum resources and other problems. In the part of spectrum prediction technology, this paper not only predicts the channel state, but also carries out more in-depth prediction experiment, and predicts the trend and amplitude of increasing spectrum signals through scarce spectrum data, which is conducive to the realization of subsequent spectrum perception, dynamic spectrum access and related cognitive radio technology.Section 1, Page 4.“Despite the high performance of LSTM, it may bring challenges in terms of computational burden and handling missing data, and since spectral data is equivalent to long time series, and LSTM has reduced prediction accuracy for long sequences, so we pro-pose B-LTF algorithm to improve the problem of long sequence prediction. It slows down the impact of sequence length change on the prediction performance of the net-work, so that the overall spectrum prediction rate is improved. Secondly, in this paper, we not only predict the channel state, but also predict the spectrum signal. The prediction of channel state is first processed by gate threshold and then predicted, so as to know the channel state of the node in the future time, and the signal spectrum prediction goes deeper into the deeper level, that is, the trend and amplitude change of the future signal can be known through the sparse spectrum signal data, which is of great help to the subsequent cognitive radio technology.”Section 4.2, Page 16.“For the spectrum signal, after processing according to the threshold value, we can know the channel usage of the current time slot, so as to predict the future state. In addition, it can also directly predict the size and trend of the future spectrum signal through the internal relationship between the current spectrum value of the spectrum signal. Channel state only predicts whether the channel is occupied, while signal spectrum prediction goes deeper and can predict the signal amplitude. The purpose of the two is not the same. If it is access, that is, as long as the idle channel is found, and the signal spectrum prediction can be used to complete the signal spectrum, that is, to complete the signal spectrum with a small amount of sparse data and prediction. Since the spectrum signal can be regarded as a time series, neural network can also be used for sequence-to-sequence prediction.”Section 4.1.4, Page 15.“As can be seen from Figure 13, when the sequence length changes, the training results of different models are different, resulting in different prediction effects. In general, the B-LTF model has the best prediction effect. It can be seen that when the sequence length is 1000, the prediction accuracy is more than 80%, and the prediction effect of LSTM model and GRU model is not much different. The BP model is lower, but the prediction rate is also about 60%. At the same time, for the same model, the prediction accuracy shows an upward trend with the increase of the total sequence length. This is due to the increase of historical information, the model can better learn and find the relationship between each point, thereby improving the prediction accuracy. However, it can be found that the growth trend will gradually become smaller, so whether the final prediction accuracy will saturate when the sequence is long enough needs to be tested with more data sets.”
Comment 4: Considering the above, the claimed contributions must be rewritten. I recommend you rewrite paragraph 3 of the article.
Response 4: Thanks for pointing out these suggestions. We updated and added related knowledge in the Section 3, which have been presented in the revised manuscript.Contributions, Page 4.“We study the problem of spectrum availability prediction and discuss the time spectrum occupancy characteristics together.We proposed the B-LTF algorithm, combined BP network with LSTM, built a new network structure, and realized the spectrum prediction from the neural network. We studied the influence of the sequence length of spectrum data and the model prediction rate in detail, and effectively improved the accuracy of spectrum prediction.We show that the analysis of simulation prediction values obtained by simulating the current channel state shows that long short-term memory network and its improved model can effectively process the time series, and the GRU model has a simpler structure, and the training time of the GRU model will be significantly reduced compared with the LSTM network, and the im-proved B-LTF algorithm compared with the LSTM, BP and GRU have better predictive performance. In addition, when the sequence length of spectrum data increases, the model prediction rate tends to be saturated or reduced.”Section 3.1, Page 10.“Therefore, according to this situation, we proposed B-LTF algorithm, which is mainly a prediction model composed of BP neural network and LSTM network. Among them, BP neural network is not only simple in structure, but also has strong nonlinear mapping ability. However, it does not consider the time correlation between the data, which often has a big problem for the prediction of time series. When facing the same data set, it will still wait for the same result after transformation. In addition, some traditional neural networks are prone to problems such as the disappearance of gradient in the prediction, but LSTM network changes this situation. It can solve the gradient problem in the prediction, and because of the existence of gates, LSTM neural network has better performance in dealing with time series problems. Therefore, combining the advantages of BP neural network and LSTM neural network, the construction of BP-LSTM combined network prediction model can not only solve the deficiencies of BP model, but also improve the convergence speed of the overall model, so as to improve the prediction rate of the model.”
Comment 5: In paragraph 4 of the article, please present more details about the experimental part.
Response 5: Thanks for pointing out these errors. We added more details in the Section 4. Some of them as follow:Section 4.1, Page 11.“In this paper, the spectrum data are from the use of high-performance spectrum analyzer AgilentE4440A by RWTH Aachen University in Germany on the wireless spectrum of a school in Maastricht, Netherlands. The measurement band range is mainly 20MHz to 1.52GHz and 1.5GHz to 3GHz. The whole measurement band is scanned each time, and the scanning time is 1.8s on average. The measurement resolution bandwidth is 200KHz. Because of the huge number of spectrum data measured, it is not conducive to the simulation of all the spectrum data, so we selected the frequency band data from 890MHz to 915MHz (GSM900 uplink band) and 935MHz to 960MHz (GSM900 downlink band) from the collected spectrum data for processing and data analysis. Then nine channels from the downlink band of GSM900 are selected as the data of this experiment for the training and prediction of the model.”Section 4.1.4, Page14.“We used the processed data in 4.1.1 for experiments to study the relationship be-tween the total sequence length and the model prediction rate. In this experiment, by changing the length of the sequence, the prediction effects of the four models under different sequence lengths are obtained, and the performance of the model prediction is judged by comparing the prediction accuracy of the spectrum signal prediction of these models. A detailed simulation of the prediction rate is shown in Figure 12”.Section 4.1.4, Page15.“In the four simulation diagrams, the horizontal coordinate is the length of the sequence, the vertical coordinate is the channel state value, the blue line represents the current real channel state, and the red line represents the channel state predicted by the four models. The more red overlaps with blue, the higher the prediction accuracy of the model.”
Comment 6: I recommend that the experimental results be presented in a single figure. Predictions (a)-(e) should be presented in a single figure (use different colors).
Response 6: Thank you for your valid suggestion! We tried to present them in a single figure by using different colors, but it would be indistinguishable, so we finally decided to divide it into four figures, each consisting of the predicted value of a model compared to the true value. The detailed figures are in the revised manuscript.
Comment 7: The conclusions must be much more relevant.
Response 7: Thanks for your valuable comments. In light of these comments, some specifics have been added in conclusion. The detailed changes are as follow:“In this paper, through the processing and analysis of an actual spectrum observation data set, we find that the spectrum utilization rate of GSM900 band is not high, and there is a lot of waste of resources, which proves that the spectrum in today's world has a major problem. Based on this, we propose the B-LTF algorithm to predict the spectrum using neural networks.”
